# Organophosphate Insecticide Toxicity in Neural Development, Cognition, Behaviour and Degeneration: Insights from Zebrafish

**DOI:** 10.3390/jdb10040049

**Published:** 2022-11-21

**Authors:** Jeremy Neylon, Jarrad N. Fuller, Chris van der Poel, Jarrod E. Church, Sebastian Dworkin

**Affiliations:** Department of Microbiology, Anatomy, Physiology and Pharmacology, La Trobe University, Melbourne, VIC 3086, Australia

**Keywords:** organophosphate, insecticides, zebrafish, neurodevelopment

## Abstract

Organophosphate (OP) insecticides are used to eliminate agricultural threats posed by insects, through inhibition of the neurotransmitter acetylcholinesterase (AChE). These potent neurotoxins are extremely efficacious in insect elimination, and as such, are the preferred agricultural insecticides worldwide. Despite their efficacy, however, estimates indicate that only 0.1% of organophosphates reach their desired target. Moreover, multiple studies have shown that OP exposure in both humans and animals can lead to aberrations in embryonic development, defects in childhood neurocognition, and substantial contribution to neurodegenerative diseases such as Alzheimer’s and Motor Neurone Disease. Here, we review the current state of knowledge pertaining to organophosphate exposure on both embryonic development and/or subsequent neurological consequences on behaviour, paying particular attention to data gleaned using an excellent animal model, the zebrafish (*Danio rerio*).

## 1. The Need for Insecticides

Insecticides are a class of pesticides used to eliminate insects and are employed in areas such as crop development, animal husbandry, agriculture and household environments [1]. On a global scale, insecticides account for 20% of the pesticide market [2], where their widespread use in agriculture improves crop quality and yield, and optimises the revenue and economic yield of harvests [3,4,5]. Additionally, the use of insecticides protects consumers against vector-borne diseases [6,7], forming a crucial front-line defence to safeguard the agricultural sector. However, these pesticides do not only eliminate the insects they target, but, as we will discuss here, also cause significant off-target effects to livestock, aquatic species and humans.

Whilst insects constitute an integral component of terrestrial ecosystems [8,9], their control requires significant expenditure from the agricultural sector. In 2012–2013, usage in Australian agriculture totalled approximately $350 billion on insecticides [10,11], demonstrating the economic burden of pests in agrarian settings. Despite this, the alternative (that is, non-use) is financially unviable owing to the limitless potential for insect-mediated agricultural destruction, indicating that insecticides are necessary to safeguard and maintain food production. With an increasing population and limited agrarian land available, improving production and productivity on existing land is integral for safeguarding long-term food security [12]. Therefore, insect management is critical in commercial agriculture, yet the dangerous neurotoxic activity of the most commonly used insecticides must be strictly modulated, in order to ensure both efficacy and safety to non-insect species.

## 2. Insecticides—The Move to Organophosphates (OPs)

Among the earliest implementation of synthetic compounds for insecticidal purposes was the development of organochlorine (OC) insecticides, such as dichlorodiphenyl-trichloroethane (DDT), cyclodienes including aldrin, dieldrin, heptachlor and chlordane [7,13]. These compounds are neurotoxic and exert their effects on target organisms by rapidly opening sodium (Na^+^) channels in neurons, which results in the continuous stimulation of cellular receptors [14,15]. Although remarkably efficacious against insects, the use of OC insecticides in agriculture is now largely banned worldwide, owing to the significant teratogenic and carcinogenic effects observed in wildlife and livestock, as well as humans. Further off-target effects including carcinogenesis, hypertrophy and reduced fertility in rodent models, as well as genotoxicity, reduced fertility and neurotoxicity [16], have dictated the rapid development of alternative insecticides.

Today, the vast majority of insecticides used are organophosphates (OP), which, although also neurotoxic, are generally considered a “safer” alternative [17], as the levels of bioaccumulation (accumulation of chemicals inside an organism through direct or indirect uptake) are far lower for OP rather than OC insecticides [18]. Moreover, OP insecticides are further favoured in agricultural settings due to their cost effectiveness [1] as well as their rapid mode of action (immediate neurotoxicity) against a wide variety of target organisms, leading to broad-spectrum success in pest elimination [19]. The most commonly used OPs are shown in Table 1, and include chemicals to safeguard livestock in both terrestrial and aquatic-based farming approaches.

## 3. OPs—Mode of Action In Vivo

The primary neurotoxic action of organophosphate (OP) insecticide exposure is the irreversible inhibition of acetylcholinesterase (AChE) in the synaptic junction of neurons (Figure 1), leading to the hyperstimulation of post-synaptic cells [7,20]. At a biochemical level, OPs bind to the hydroxyl group of AChE through phosphorylation, preventing AChE from hydrolysing ACh [21]. In turn, the reduction of AChE activity results in an abnormal build-up of ACh in the synaptic junctions, leading to hyperstimulation of the muscarinic and nicotinic receptors involved in cholinergic pathways [22]. Therefore, increased OP exposure is likely to substantially impact upon neural function, and the risk of OP-dependent toxicity can be quantitated through measuring AChE levels and activity in blood [23] and pesticide metabolites, such as dialkyl phosphate (DAP) compounds in urine [24,25] and plasma [26].

AChE is integral for regulating neurotransmission [27], achieved through the hydrolysing of acetylcholine (ACh) into its two structural products, acetyl CoA and choline, the latter of which is returned to the pre-synaptic neuron via the sodium (Na^+^) choline transporter, enabling regeneration of ACh in the pre-synaptic neuron. This process occurs in the central nervous system (CNS) within the brainstem, striatum and basal forebrain [28], as well as in the peripheral nervous system (PNS) at both the neuroglandular and neuromuscular junctions [29]. ACh from pre-synaptic neurons plays important roles in initiating skeletal, smooth and cardiac muscle contraction, and more recently have been implicated in the development and maturation of the brain [30,31,32].

As a result of acute and chronic exposure, OPs can cause extensive damage to cells including cytotoxicity, apoptosis [33,34], genotoxicity and subsequent DNA mutations [35]. In previous studies, pesticides have been shown to produce covalent adducts with DNA, and as a result form interstrand cross-links which inhibit cellular replication and transcription [36]. Given that DNA adducts are destructive compounds that cause cellular damage, they are nonetheless useful biomarkers for identifying oxidative stress and genotoxicity [37].

Other cellular irregularities identified include increases in cytokine secretion (particularly tumour necrosis factor and interleukin 6) through neuroinflammation, improper clearance of reactive oxygen species, and pathologically-induced alterations in gene expression [38,39,40].

## 4. OPs—Occupational, Household and Waterway Exposure

Although OP insecticides are highly effective, it is estimated that only 0.1% of these pesticides reach their target organisms [41,42], with the majority being lost in soil, food and drainage [22,43,44]; in fact, OPs are the most common synthetic material found in waterways, soil and animal tissues [5]. OP exposure is generally considered to be most problematic in agricultural environments, and indeed occupational settings, such as farms and factories, contribute a significant avenue of OP exposure. However, OP exposure frequently occurs also in the household environment through food contamination and oral/epidermal exposure [27]. Although less common than occupational exposure, residential OP exposure can occur from the use of these insecticides in the household or garden (Figure 2), and usually occur from improper storage and spills [45]. Additionally, the infiltration of OPs in the diet, where OPs are consistently detected in foods at low levels [46], is considered to be a significant route of exposure and subsequent poisonings [47]. Children are particularly vulnerable to OP exposure through diet due to the fact that they eat 2.8–4.8 times more food per unit of body mass, and the types of food that they eat (fruits and vegetables) contain higher levels of OP residues [47,48]. Although techniques do exist for monitoring environmental OP exposure (surrogate skin, fluorescent tracers, air sampling pumps, etc.), most OPs are only detected once they have entered the body [49]. Therefore, greater understanding of the lifelong effects of OP exposure are necessary in order to better govern their use and mandate appropriate safety measures for use where the likelihood of human ingestion is high.

Waterway exposure can occur unintentionally via agricultural surface run-off, creating a significant risk not only directly to aquatic ecosystems, but indirectly to humans as well [50]. OPs can also be intentionally released into waterways, a phenomenon commonly observed in commercial fish farms and fishing sports that use these substances to eliminate water-borne pests [51] such as flat worm parasites [52]. Assessing chronic OP exposure in waterways is challenging owing to high OP solubility, relatively short half-lives, and relatively low bioaccumulation [53]; however, data suggest that OP contamination is of global concern. With 95% of urban streams in the US showing detectable levels of OP contamination [54], one can assume that aquatic species are consistently exposed to OP insecticides for prolonged periods.

## 5. Major Findings: The Zebrafish as Model for Testing Organophosphate (OP) Insecticides

The use of mammalian models such as mice for screening experiments is both expensive and, owing to the relatively lengthy gestation periods, time consuming. Therefore, alternate animal models of environmental susceptibility, which are low cost and high-throughput, are desirable for toxin-screening, as these would provide rapid functional data that could narrow down OPs of interest and allow for subsequent targeted testing in mammalian models.

One such model is the zebrafish (*Danio rerio*), which has become a favoured model for developmental research [55]. The zebrafish is an excellent model of human diseases, as the zebrafish and human genome share more than ~80% similarity [56]. This well-established genetic conservation of the zebrafish is one of the reasons why it is supported as a model for environmental toxicology studies, specifically in relation to vertebrate embryogenesis [57]. The rapid rate in which the structures of the zebrafish develop, coupled with optical clarity and ease of access, makes it a model of choice for observing embryo development.

The toxic effects of OP insecticides at early developmental stages of zebrafish embryogenesis have been investigated in multiple studies, which for the first time we have collated here (Table 2). From an embryological toxicity standpoint, a substantial number of these studies employed an acute exposure period from 0–5 h post-fertilization (hpf), allowing early phenotypes to be investigated. These timepoints coincide with the initial critical stages of cellular proliferation, migration (epiboly) and the onset of gastrulation, processes essential for the establishment of the three germinal layers and subsequent patterning of tissue primordia. The consequences of OP exposure in these acute early-stage studies are diverse; however, various morphological (spinal, yolk sac, body length, pigment and eye surface area), physiological (heart rate, AChE levels, genetic), and behavioural (locomotor activity, anxiety) impairments are commonly identified. Exposure to OPs at zebrafish adult stages led to predominately behavioural (anxiety, startle response) and physiological (ATP, AChE, GSH, MDA, etc.) irregularities, with fewer concomitant morphological impairments, highly consistent with lifelong morbidity as a consequence of acute exposure primarily in the early stages of development.

While symptoms and defects present in zebrafish models do not always accurately predict human disease, the broad effects of OP on development appear consistent across both fish and humans. The neurological effects of OP in development appear largely consistent across species and correlate with the known mechanism of action of OP on the cholinergic system. Further, the >80% commonality in genome between zebrafish and humans indicates that zebrafish studies are valuable for identifying the molecular changes that may be common to harmful OP exposure in humans and zebrafish. To this end, we believe our summary table will serve as an invaluable resource for future continued implementation of the zebrafish in determining consequences for human OP-dependent disease, as indicated in the sections below.

## 6. Organophosphate Toxicity—Acute Cholinergic Syndrome (ACS)

Although having conserved function as a neurotoxin, the phenotypic consequences of OP exposure are nonetheless extremely variable, and symptoms may manifest following either acute (high dose) or chronic (typically lower dose) exposure (Figure 3). The earliest stage of OP toxicity is referred to as acute cholinergic syndrome (ACS), which is a result of the effects of AChE inhibition [93]. ACS can occur within minutes of OP exposure, and impairs both muscarinic and nicotinic receptors found in the nervous system [94]. The consequences of hyper-stimulated post-synaptic receptors (hyperstimulation) vary depending on their locations. In the CNS, hyperstimulation more commonly occurs at muscarinic receptors, resulting in heart irregularities, gastrointestinal issues including stomach cramps, diarrhoea and vomiting, respiratory complications including bronchorrhea and bronchospasms, as well as neurological effects including seizures, agitations and anxiety [95,96]. Comparatively, hyperstimulation in the PNS is more commonly associated with nicotinic receptors, and this is often expressed as muscle weakness, cramps or paralysis [7,97]. Symptoms of OP exposure are quite complex in that they are not limited to a localised area and are a result of the diverse effect of OPs on both CNS and PNS pathways.

## 7. Organophosphate Toxicity—Intermediate Syndrome (IMS)

The intermediate syndrome (IMS) follows 1–4 days after ACS (Figure 4), and is characterised as the onset of muscle weakness, particularly in the proximal limbs, neck and respiratory system [98]. If untreated, from 14–21 days after acute exposure, weakness in the peripheral muscles becomes evident [49]. It is estimated that only ~20% of humans exposed to OP will have symptoms that progress from ACS to the IMS stage [99]. The IMS is commonly associated with respiratory failure as a result of nicotinic receptor paralysis. Respiratory failure from prolonged OP exposure is primarily linked to the CNS, specifically the depression of the pre-Botzinger complex (glutaminergic and muscarinic fibres) located in the ventrolateral medulla in the brainstem [100]. This has resulted in respiratory failure being recognised as a significant comorbidity in OP mortality [94].

## 8. Organophosphate Toxicity—Organophosphate-Induced Delayed Neuropathy (OPIDN)

OP toxicity does not solely result in AChE inhibition, and depending on the physical structure of the compound, OPs can target other secondary hydroxyl sites on enzymes other than AChE [101]. OP insecticide exposure can also give rise to another type of toxicity, referred to as organophosphate-induced delayed neuropathy (OPIDN), which, depending on dose and chemical structure, occurs ~2–3 weeks after ACS [94,102]. This pathology is characterised as the degeneration of distal axons in the CNS and PNS, is expressed as sensory loss in both hands and feet, weakness in distal muscles and coordination issues [7] and is associated with the inhibition of neuropathy-target-esterase (NTE) [103]. NTE is an integral enzyme employed at the neurite initiation stage of neuronal morphogenesis, with these neurites maturing into axons and dendrites to form part of the nervous system [104].

Importantly, the consequences of OPIDN are only present when ≥70% of NTE is inhibited [105]. Additionally, in order for irreversible inhibition of NTE, there must be a secondary chemical reaction, where there is a displacement of an R-group (aging) [103], and, therefore, NTE inhibition and OPIDN is thought to contribute to diseases characterised by axonal degradation such as Alzheimer’s disease, Parkinson’s disease and motor neuron diseases (MND) that include amyotrophic lateral sclerosis (ALS) and progressive bulbar palsy [106]. As these diseases are primarily associated with aging in humans, it is interesting to note that animal models show that adults are both far more susceptible to, and recover far more poorly from, OPIDN than juveniles [106].

The inhibition of NTE itself is not responsible for axonal degeneration, as has been demonstrated with non-OP inhibitors (organophosphinates, sulfonyl fluorides and carbamates) that covalently react with NTE, but do not undergo the enzyme ageing process [107]; this indicates that R-group displacement confers a “gain” of neurotoxicity that is damaging in its own right.

## 9. Organophosphate Toxicity—Effects on Embryogenesis

Pre-natal OP exposure is of particular concern, as developing babies are highly susceptible to chemical injury [108,109]. This is partly a result of their immature detoxification mechanisms, i.e., reduced expression of OP specific detoxifying enzymes such as paraoxonase and chlorpyrifos-oxonase, compared to adults [110,111], and also as the cholinergic system (which is targeted by OPs) is heavily involved with placental processes including amino acid uptake and nitric oxide signalling [112]. Pre-natal OP exposure has been associated with shortened gestational periods [113], reduced birth weight and birth length [114], as well as impaired reflexes [115] and neurobehavioral irregularities [116].

While the influence of OPs on the mature blood brain barrier (BBB) is unclear, the developing and newborn BBB is “leaky”, allowing toxins, in particularly pesticides in the fetal circulation, to cross and have negative effects on the developing brain [117,118]. Overall, the neurotoxic properties of OPs, and the resultant syndromes have been reasonably well documented (Figure 4), with AChE inhibition and hyperstimulation of postsynaptic neurons being key contributors to these impairments [119].

## 10. Organophosphate Toxicity—Effects on Neurodevelopment and Early Behaviour

The timing of prenatal OP exposure plays a critical role in fetal development and postnatal behaviour. OP exposure during the 1st and 2nd trimester of pregnancy has been shown to be associated with delayed cognitive performance at both 2 and 6 months of age, whereas OP exposure in the 3rd trimester of pregnancy is associated with delayed communication and motor performance at 6 months of age [120].

In terms of neurodevelopmental impairments, OPs such as chlorpyrifos and diazinon have been shown to cause decreased DNA synthesis in neuronotypic PC12 and gliotypic C6 neural cell lines, the latter of which continues to develop into the postnatal period [121]. Additionally, children aged ~3 years have been identified as having increased risk of displaying developmental delays and a higher incidence of behavioural disorders such as ADHD [108,122], and prenatal exposure to OP insecticides was associated with poorer intellectual development in seven year old children [123], as well as poorer motor skills and cognitive recall when compared to non-exposed children [124]. Taken together, these studies show that not only does pre-natal exposure to OPs affect embryogenesis, but also cognitive development in young children.

## 11. Organophosphate Toxicity—Effects in Adulthood and Neurodegenerative Diseases

Whilst infants and children are highly susceptible to OP insecticide toxicity, these chemicals have also been associated with impaired health at later stages of life, particularly neurological disorders such as Alzheimer’s Disease (AD) and Parkinson’s Disease (PD) [125]. Recent studies have shown that chronic exposure in agricultural workers is associated with neural irregularities including neurodegenerative diseases, attention impairment and short-term memory loss [11]. The cholinergic system, which is affected by OP insecticides, has long been associated with neurodegenerative diseases, with ACh one of the key neurotransmitters involved in cellular signalling in the brain [125]. The reduction of ACh is a critical element in memory loss diseases such as AD, where ACh in the basal forebrain is known to play an integral role in memory and learning; as OP insecticides promote an imbalance of ACh at cellular junctions in the brain, these chemicals are, therefore, linked to impaired memory diseases [126].

Although PD involves a depletion of dopaminergic cell bodies, it is symptomatically dissimilar to AD in that it is characterised by motor impairments such as tremor at early onset and posture/gait issues at later stages [127]. Along with genetic predisposition, pesticides are widely acknowledged as an environmental risk factor for PD, with OPs having been implicated in some studies of the disease [128,129,130]. Variability in the *PON1* gene (when exposed to various OPs—diazinon/chlorpyrifos) has been shown to correlate with a greater than two-fold increase in PD risk [131]. Despite the causal relationship being still largely unknown, an epidemiological analysis of 23 case-control studies found that 13 of the studies reported a statistically significant risk of PD with pesticide exposure, with both chlorpyrifos (OP) and organochlorines (OC) being key contributors to the study [132]. However, one limitation of pesticide research on neurodegenerative diseases such as PD is that the study of pesticides does not encompass the lifespan, making it difficult to analyse the long term effects of these chemicals [133].

## 12. Conclusions

Rapid population growth and changing diets in developing countries have increased the demand for food, to the point that food production must increase by 70% to meet the estimated food demands in 2050 [134]. Pesticides will play an essential role in achieving this production increase. While the economic and societal benefits of pesticides are inarguable, the effects of pesticide exposure on non-target animals and humans are a continuing concern. Of note is the growing use of OP insecticides, which are linked to poor neurodevelopment in both developed and developing countries worldwide. While our understanding of the relationship between OP exposure and poor health outcomes is growing, it remains unclear the extent to which AChE inhibition and OP exposure lead to developmental abnormalities. However, addressing this relationship between OP exposure and neurodevelopment and behaviour will encourage improvements in the regulation of use and handling of OPs in agricultural, industrial and domestic environments around the world.

## Figures and Tables

**Figure 1 jdb-10-00049-f001:**
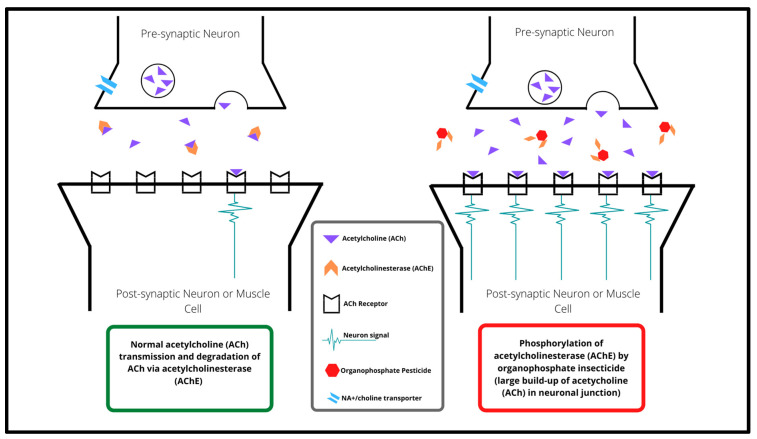
The physiological action of acetylcholine (ACh; purple triangles) at the neuronal cell synapse, the breakdown of ACh through acetylcholinesterase (AChE; orange diamonds), and the phosphorylation of AChE through organophosphate insecticide (OP; red hexagons) exposure.

**Figure 2 jdb-10-00049-f002:**
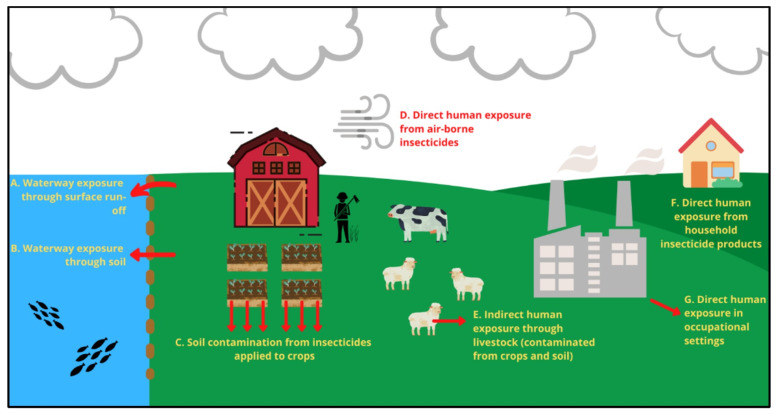
The primary direct and indirect routes of organophosphate (OP) exposure on target and non-target organisms in agricultural, household and aquatic environments.

**Figure 3 jdb-10-00049-f003:**
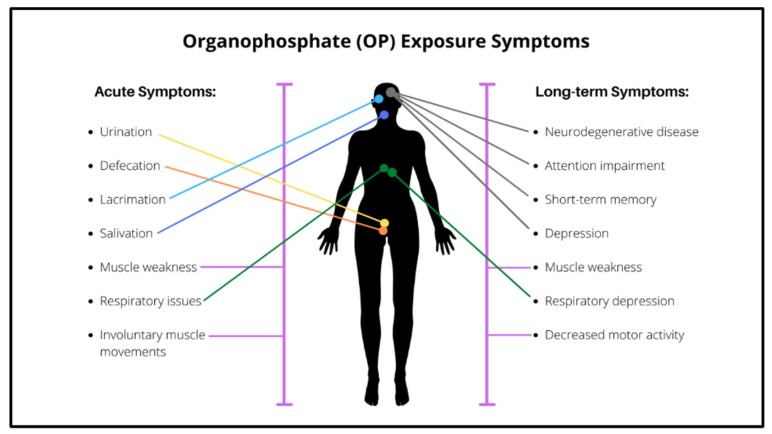
The major consequences of OP exposure in humans.

**Figure 4 jdb-10-00049-f004:**
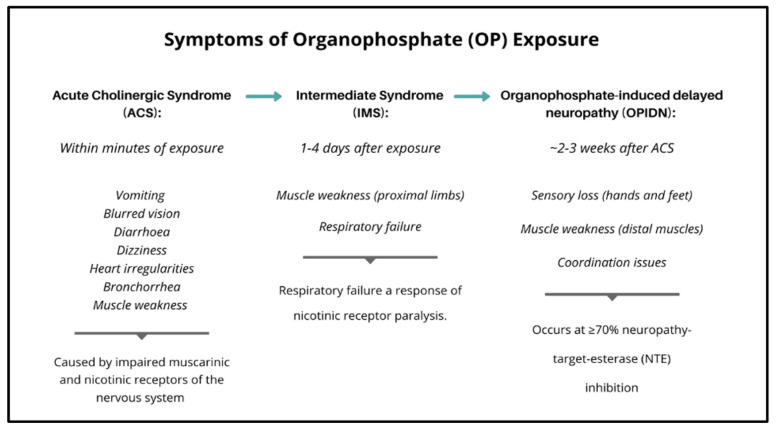
The symptoms of organophosphate (OP) exposure at progressive time points; acute cholinergic syndrome (ACS) occurs within minutes of OP exposure, symptoms of intermediate syndrome (IMS) display at 1–4 days after OP exposure, and symptoms of organophosphate-induced delayed neuropathy (OPIDN) occur ~2–3 weeks after OP exposure.

**Table 1 jdb-10-00049-t001:** Commonly used organophosphate (OP) insecticides in agriculture and aquaculture. Shown here are approvals using the Public Chemical Registration Information System Search (PubCRIS), molecular formula and general uses.

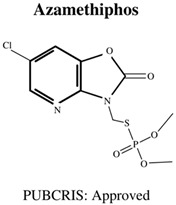	**Molecular Formula**: C_9_H_10_CIN_2_O_5_PS **General Uses of OP**: Used in Aquatic Farming (Atlantic Salmon) to Control Parasites.	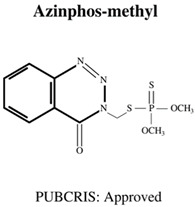	**Molecular Formula**: C_10_H_12_N_3_O_3_PS_2_ **General Uses of OP**: Used on Orchard Fruits and Nut Crops to Control Moths.
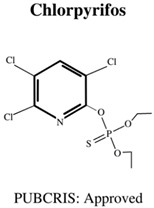	**Molecular Formula**: C_9_H_11_Cl_3_NO_3_PS **General Uses of OP**: Used broadly (crops/animals/buildings) to control roundworms, mosquitos and termites.	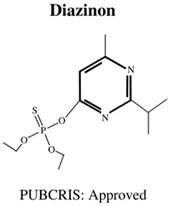	**Molecular Formula**: C_12_H_21_N_2_O_3_PS **General Uses of OP**: Used on crops (fruits/vegetables/nuts/field crops) to control ants, fleas and cockroaches.
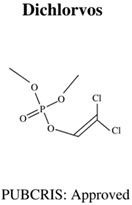	**Molecular Formula**: C_4_H_7_Cl_2_O_4_P **General Uses of OP**: Used broadly (household/ agriculture) to control flies, caterpillars, thrips and mites.	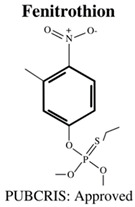	**Molecular Formula**: C_9_H_12_NO_5_PS **General Uses of OP**: Used broadly (public health/agriculture) to control beetles, grubs, locusts, flies, mosquitos, etc.
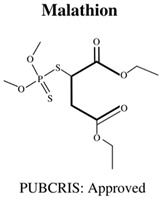	**Molecular Formula**: C_10_H_19_O_6_PS_2_ **General Uses of OP**: Used broadly (landscaping/public health/agriculture) to control mosquitos, fleas and ants.	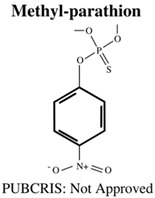	**Molecular Formula**: C_8_H_10_NO_5_PS **General Uses of OP**: Used in open fields (cotton, soybean, vegetable) to control boll weevils, etc.
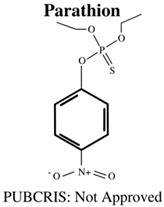	**Molecular Formula**: C_10_H_14_NO_5_PS **General Uses of OP**: No longer used (banned largely worldwide) due to its high toxicity.	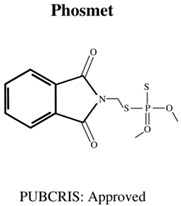	**Molecular Formula**: C_11_H_12_NO_4_PS_2_ **General Uses of OP**: Used broadly (plants/animals) to control moths, mites, flies and aphids.
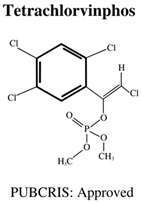	**Molecular Formula**: C_10_H_9_Cl_4_O_4_P **General Uses of OP**: Used on animals (cattle, hogs, goats, chickens, and horses) to control flies and mites.		

**Table 2 jdb-10-00049-t002:** Studies utilising zebrafish (*Danio rerio*) to determine developmental and neurotoxic effects of organophosphate insecticides. dpf; days post-fertilisation. hpf; hours post-fertilisation.

Organophosphate(s)	Dosage(s) *	Gene(s) Involved **	Exposure Period	Observations	Reference
Chlorpyrifos (CPF)	CPF: 10 & 100 ng/mL	-	0–5 dpf	- increased impaired spatial discrimination - both decreased (10 ng/mL) and increased (100 ng/mL) response latency in adult zebrafish	[58]
Chlorpyrifos (CPF)	CPF: 100 ng/mL	-	0–5 dpf	- decreased swimming activity	[59]
Malathion (MAL)	MAL: 2.5 & 3 mg/L	-	3 hpf–5 dpf	- increased mortality - decreased hatching rates - decreased body length - and decreased surface area of eye.	[60]
Malathion (MAL)	MAL: 0.25, 0.5, 1, 3 & 5 mM	-	Adult (sexually mature)	- decreased Adenosine Di-Phosphate (ADP) and Adenosine Tri-Phosphate (ATP) levels	[61]
Diazinon (DZN)	DZN: 2000 & 3000 μg/L	-	8 hpf–96 hpf	- increased heart rate - increased mortality - increased morphological irregularities (axial and tail deformities, yolk sac/heart oedema, eye irregularities - reduced pigmentation - decreased hatching rate	[62]
Chlorpyrifos (CPF)	CPF: 0.25, 0.5, 0.75 & 1 mg/L	-	Acute: 5 dpf for 2 h Sub-chronic: ≤1 hpf–11dpf	- Acute—increased (≥0.25 mg/L) and decreased (≥0.75 mg/L) locomotor activity. - Sub-chronic—increased behavioural irregularities	[63]
Chlorpyrifos (CPF)	CPF: 300, 1500 & 3000 nM	Rohon-Beard Development/Axonogenesis: *agrin↓*, *cntn2↓*, *ntf3↓*, *sema3d↓*	3 hpf–27 hpf/51 hpf/72 hpf/4 dpf	- increased mortality - increased phenotypes (axial curvature, reduced body size and reduced pigmentation) - decreased functioning AChE - increased average chevron angle (somites) - decreased HNK-1-positive cells - decreased axonogenesis-related genes	[64]
Chlorpyrifos (CPF)	CPF: 0.29 μM	-	0–5 dpf	- significantly increased startle response - increased transmitter turnover in larvae - decreased dopamine/serotonin levels in adults	[65]
Chlorpyrifos-oxon (CPF metabolite)	CPF: 300 nM	0.1 μg/L, 3 μg/L	3 hpf–75 hpf	- Defective peripheral neuron development	[66]
Dichlorvos (DCV)	DCV: 20.81, 25 & 66.78 mg/L	-	0 hpf–96 hpf	- decreased hatching rates - increased pericardial oedema - increased spinal irregularities - decreased swimming activity	[67]
Dichlorvos (DCV) Phoxim (PHO)	DCV: N/A PHO: 0.469, 0.513, 0.700 & 1.28 mg/L	-	Adult (sexually mature)	- DCV: - low toxicity (determined by LC_50_) - PHO: - intermediate and high levels of toxicity (determined by LC_50_)	[4]
Chlorpyrifos (CPF)	CPF: 0.6 μM	-	1 ypf for 24 h	- decreased swim rates - increased freeze response - decreased AChE in muscle	[68]
Chlorpyrifos (CPF)	CPF: 0.01, 0.1 & 1 μM	-	6 hpf–24/48/72 hpf	- decreased functioning AChE - increased TCPy (trichloro-2-pyridinol) - decreased functioning primary/secondary motor neurons, axonal growth and sensory neurons.	[69]
Chlorpyrifos (CPF) Diazinon (DZN) Parathion (PA)	CPF: 0.3, 3 & 30 μM DZN: 10 & 30 μM PA: 10 & 30 μM	-	6 hpf–5 dpf	- ***Both CPF and DZN***: - increased mortality - decreased functioning AChE - decreased locomotor activity - ***PA***: - increased mortality - decreased functioning AChE	[70]
Chlorpyrifos (CPF)	CPF: 0.01 & 0.1 μM	-	0–7 dpf	- decreased swim speed - decreased anxiety-like behaviour - increased behavioural irregularities - shortened body lengths and tail defects	[71]
Monocrotophos (MCP)	MCP: 0.001 & 0.100 mg/L	Sexual Differentiation: *cyp19a1a↑*, *cyp19a1b↑*, *foxl2↑*, *dmrt1↓*, *B-actin*, *ef1-a*	72 hpfV–16 dpf	- increased proportion of females - alteration in expression of sexual differentiation genes	[72]
Chlorpyrifos (CPF) Dichlorvos (DCV) Diazinon (DZN)	CPF: 1, 10, 100 & 1000 μM DCV: 100 & 1000 μM DZN: 100 & 1000 μM	-	1 hpf–5 dpf	- CPF: - increased mortality - increased kyphosis - decreased spine length - increased spontaneous movement - and decreased heart rate - DCV: - increased mortality - increased spontaneous movement. - DZN: - increased mortality - increased pericardial oedema	[73]
Chlorpyrifos (CPF)	CPF: 30, 100 & 300 μg/L	*Gfap*, *Mbp↓*, *Elavl3↑*, *Ngn1↑*, *Nestin↑*, *Shha↑*	0–5 dpf	- significantly decreased hatching rates - increased spine and yolk sac abnormalities - significantly decreased heart rates - significantly decreased swim speed/distance	[50]
Chlorpyrifos (CPF)	CPF: 200 & 400 μg/L	-	2 hpfV–72 hpf	- decreased AChE activity - increased AChE gene expression - increased glutathione (GSH) levels	[74]
Dichlorvos (DCV)	DCV: 6, 19, & 32 mg/L	Oxidative Stress: *Nrf2* (many other associated genes within the Nrf2 pathway also examined)	6–12 mpf	- decreased cholinesterase (ChE) levels in the heart/brain - increased myo-degeneration - increased testis degeneration - increased pancreas zymogen granule depletion - decreased glycogen in liver - altered expression of genes involved in Nrf2 signalling	[75]
Monocrotophos (MCP)	MCP: 10, 20, 30, 40, 50 & 60 mg/L	-	4 hpf–96 hpf	- moderate toxicity (determined by LC_50_) - decreased body length - decreased heart rate - decreased functioning AChE levels	[1]
Monocrotophos (MCP)	MCP: 100 μg/L	HPI Axis: *Crf*, *Gr↓*, *POMC↓*, *P450_11β_*, *11B-HSD2*, *StAR*, *20B-HSD2↑*, *MC2R↓*, *TAT*, *PEPCK*	Adult (sexually mature)—21 d exposure	- decreased whole-body cortisol - increased/decreased hypothalamic-pituitary-inter-renal (HPI) axis associated genes	[76]
Chlorpyrifos (CPF)	CPF: 2 & 5 μM	-	Adult (sexually mature)	- increased oxidative stress - decreased neurotransmitter metabolism - increased energy exhaustion	[77]
Chlorpyrifos (CPF) Phoxim (PHO)	CPF: 0.28- 13.03 mg/L PHO: 0.89–26.48 mg/L	-	Embryo (1 hpf), larvae (72 hpf) and juvenile (1 mpf)—96 h exposure	- CPF was determined to be more toxic than PHO (determined by LC_50_)	[78]
Diazinon (DZN)	DZN: 6.5 mg/L	-	6 hpf–5 dpf	- Moderate toxicity (determined by LC_50_)	[2]
Dichlorvos (DCV)	DCV: 15 mg/L	-	Adult (sexually mature) 4–5 m—24 h exposure	- increased levels of malondialdehyde (MDA) in liver/kidney - increased glutathione (GSH) in liver/kidney/brain - increased superoxide dismutase in liver - decreased levels of superoxide dismutase in brain - decreased catalase in kidney/brain.	[79]
Malathion (MAL)	MAL: 250, 500 & 1000 μg/L	HPG Axis: *vtg1*, *vtg2*, *era↑*, *erB1*, *erB2*, *cyp19a1a*, *cyp19a1b↑*	6 dpf–10 dpf	- low toxicity (determined by LC_50_) - upregulation of gene expression within the hypothalamic-pituitary-gonadal (HPG) axis	[80]
Chlorpyrifos (CPF) Diazinon (DZN)	CPF: 1, 10, & 25 μM DZN: 10 & 100 μM	-	6 hpf–102 hpf	- CPF: - increased mortality - decreased hatching rates - increased spinal lordosis - reduced activity - DZN: - increased mortality - increased pericardial oedema - decreased mitochondrial bioenergetics	[18]
Monocrotophos (MCP)	MCP: 0.125, 0.625 & 1.25 uL/L		24–72 hpf	- DNA damage observed in peripheral blood	[81]
Phosalone (PSL)	PSL: 86–505 μg/L	-	8 wpf–96 h exposure	- decreased functioning AChE - decreased carboxylesterase (CaE) - increased glutathione (GSH)	[82]
Chlorpyrifos (CPF)	CPF: 30, 100 & 300 μg/L	Oxidative stress: *Mn-Sod↑/↓*, *Cu/Zn-Sod↓*, *Gpx↓*, *Cat*↓, *Ucp2*↓, *bc12*, *Cox1*↓ Glycolysis/Lipid: *Gk*↓, *HK1*, *Pk*↓, *Pepckc*↓, *Aco*↓, *CPt1*↓, *Ppar-A*↓, *Acc1*↓, *Srebp 1a*↓, *Ppar-y*↓, *Fas*↓, *Fabp6*, *Apo*↓, *Dgat*↓, *LDLR*↓, *HMGCR*, *Fabp5*	Adult (sexually mature)	- increased levels of gut mucus - decreased y-Protobacteria in gut - decreased oxidative stress genes in gut and liver - and decreased glycolysis and lipid metabolism-related genes	[56]
Chlorpyrifos (CPF)	CPF: 30, 100 & 300 μg/L	Cardiovascular: *Mef2c*↓, *Bmp4*↓, *VEGFR-2*, *JunB↑*, *Tbx2* Lipid: *Ppar-a*, *Ppar-y*↓, *Srebp 1a*, *Acc1*, *Fas*↓, *Cpt1*↓, *Aco*, *Apo*↓, *Fabp5*, *Fabp6*↓, *Dgat*↓, *LDLR*	2 hpf–7 dpf	- decreased lipid accumulation in heart - decreased triglyceride and total cholesterol - increased cellular apoptosis of heart tissue - decreased lipid metabolism genes	[83]
Diazinon (DZN) Dichlorvos (DCV) Malathion (MAL) Parathion (PA)	DZN: 0.1 & 100 μg/L DCV: N/A MAL: 100 μg/L PA: 0.1 μg/L	Cholinergic: *AChE↑/↓* Neurodegeneration: *c-Fos*, *lingo-1b↑*, *grin-1b↓*	5 hpf–5 dpf	- DZN: - decreased swimming distance - decreased velocity - increase in AChE associated gene expression inhibited functioning AChe - increased carboxylesterase activity - DCV: - increased AChE associated genes. - MAL: - decreased swimming distance - decreased velocity - increase in AChE associated gene expression - increase in neurodegenerative associated gene expression - increased carboxylesterase activity - PA: - decrease in AChE associated gene expression - and decrease in neurodegenerative associated gene expression	[84]
Dichlorvos (DCV)	DCV: 1, 5 & 10 mg/L	-	1 hpf–7 dpf	- decreased body length - decreased heart rates - decreased surface area of eye - decreased escape responses - decreased speed - decreased mobile time	[85]
Sumithion (SMT)	SMT: 1 mg/L	-	Adult (sexually mature)—96h exposure	- increased blood glucose levels - increased frequency of micronucleus in erythrocytes - increased erythrocyte cellular and nuclear abnormalities	[86]
Chlorpyrifos (CPF)	CPF: 1 & 3 μM	-	Adult (sexually mature) 6–8 m—2/5 w exposure	- increased anxiety related activity (Novel Tank Diving Test) - increased approach response in shoaling assay - increased predator avoidance activity (predator avoidance assay)	[87,88]
Chlorpyrifos (CPF) Malathion (MAL)	CPF: 0.019, 0.077, 0.31, 0.41, 1.01, 1.53 & 6.15 mg/L MAL: 0.039, 0.16, 0.62, 2.90, 8.04, 8.54 & 12.45 mg/L	Oxidative Stress: *Cat*, *CuSod*, *MnSod* Immunity: *Cxcl↓*, *IL↑*, *Tnf↑/↓* Apoptosis: *Cas8↑/↓*, *Cas9*, *P53*, *Bax* Endocrine: *TRa*, *TRb↓*, *ERa*, *Tsh↓*, *Crh*, *cyp19a↑*	1 hpf–96 hpf	- Both CPF and MAL: - severe toxicity at larvae, juvenile and adult stages (compared to embryo stage) - significant changes in expression of immunity, apoptosis, and endocrine related genes	[57]
Sumithion (SMT)	SMT: 0.1, 0.2, 0.4, 0.8 & 1.6 mg/L	-	Embryo/larvae	- increased mortality in embryos and larvae - decreased hatching rates - increased morphological irregularities in embryos (damaged/underdeveloped and darkened yolk sac, broken chorion, and aberrant notochord formation) - increased morphological irregularities in larvae (yolk sac ulcerations/swelling and oedema, heart damage, lesion at caudal region, uninflated swim bladder, head malformation, jaw irregularities, and notochord abnormalities).	[89]
Chlorpyrifos (CPF)	CPF: 1μM	-	Adult (sexually mature)—5 w exposure	- decreased brain cholinesterase (ChE) activity - increased fleeing score.	(Hawkey, 2021)
Diazinon (DZN)	DZN: 0.4, 1.25 & 4.0 μM		5–120 hpf	- Changes in Mitochondrial oxygen utilization in the brain and testes	[90]
Malathion (MAL) Chlorpyrifos (CPF)	MAL: 5, 50 ug/L CPF: 0.1 & 3 ug/L		0–14 dpf	- increases in reactive oxygen species - induction of oxidative stress	[91]
Chlorpyrifos (CPF)		*Caspase 3↓*, *Bcl-2↓*,	Adult—8–12 months old 14 day exposure	- Significantly elevated ROS levels - Elevated Reactive nitrogen species levels in high CPF dosage groups	[92]

* All dosages listed are associated with the observations summarised here, other dosages in the individual studies may have been used, but did not impact on development, behaviour or gene expression. ** ↑ and ↓ arrows indicate where there has been a significant increase or decrease in a particular gene as a response of organophosphate (OP) exposure—where no arrow shows, no significant change was noted.

## Data Availability

Not Applicable.

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
