# Peer review of "Organophosphate Insecticide Toxicity in Neural Development, Cognition, Behaviour and Degeneration: Insights from Zebrafish"

_jdb, 2022, doi:10.3390/jdb10040049_

Round 1

Reviewer 1 Report

The paper “Organophosphate insecticide toxicity in neural development, cognition, behaviour and degeneration: insights from zebrafish” is an interesting review regarding the toxicity of organophosphate insecticides with several studies using the zebrafish model. For the publication, I suggest a minor revision. The authors should revise the manuscript considering that:

- The references should be specific and indicate the bibliographical reference directly and not through citations by other papers. This revision is particularly necessary for the introduction paragraph but also for:

a) Page 4, line 103:

“Although OP insecticides are highly effective, it is estimated that only 0.1% of these pesticides reach their target organisms, with the majority being lost in soil, food and drainage [15, 31, 32]”

This sentence is much important and it is necessary to report the specific paper.

b) Page 7:

Invite to control reference 52 for Chlorpyrifos (CPF) in the table.

b) Page 4, line 88. This sentence: “ACh from pre-synaptic neurons plays important roles in initiating skeletal, smooth and cardiac muscles, influencing the release of hormones, and more recently been implicated in the development and maturation of the brain.” should be clarified and insert the bibliographic reference for “more recently been implicated in the development and maturation of the brain.”

These are some examples to explain the necessary revision that I suggest to improve the manuscript.

Other suggestions:

- Page 1, line 31: "Whilst insecticides are an integral component of terrestrial ecosystems "

What intended the authors explain? Are the insecticides normally present in terrestrial ecosystems?

- In figure 1 for organophosphate insecticide (OP; red hexagons), OP is not reported.

- The conclusion is complex to read. I suggest revising the long sentences and correcting grammatical errors.

Author Response

The paper “Organophosphate insecticide toxicity in neural development, cognition, behaviour and degeneration: insights from zebrafish” is an interesting review regarding the toxicity of organophosphate insecticides with several studies using the zebrafish model. For the publication, I suggest a minor revision. The authors should revise the manuscript considering that:

- The references should be specific and indicate the bibliographical reference directly and not through citations by other papers.

We agree that in multiple instances the references included in this review should have indeed been sourced from primary data, and as such we have replaced review articles with the original sources as requested.

The examples highlighted by the reviewer (including Naughton and Terry, 2018, Richardson  et al, 2019, Scutter, 2017, Perry  et al 2020 and Karami-Mohajeri and Abdollahi, 2011) have either been replaced, or supplemented with the original data papers.

This revision is particularly necessary for the introduction paragraph but also for:

  1. a) Page 4, line 103:

“Although OP insecticides are highly effective, it is estimated that only 0.1% of these pesticides reach their target organisms, with the majority being lost in soil, food and drainage [15, 31, 32]”

This sentence is much important and it is necessary to report the specific paper.

We agree, and have included the original study (Pimentel and Levin; 1992) alongside a follow-up 1995 study (Pimentel, 1995).

  1. b) Page 7:

Invite to control reference 52 for Chlorpyrifos (CPF) in the table.

With regard to reference 52 for CPF in the pesticides table, it is not clear what this comment refers to; most, if not all, primary data papers investigating chlorpyrifos in zebrafish have been referenced in the table. Is the reviewer alluding to a specific reference, that we have inadvertently missed? To the best of our knowledge, the original study examining chlorpyrifos in zebrafish development was the 2003 study of Levin et. al., Neurotoxicol Teratol (Levin, 2003), that we have cited (updated reference #58).

  1. b) Page 4, line 88. This sentence: “ACh from pre-synaptic neurons plays important roles in initiating skeletal, smooth and cardiac muscles, influencing the release of hormones, and more recently been implicated in the development and maturation of the brain.” should be clarified and insert the bibliographic reference for “more recently been implicated in the development and maturation of the brain.”

We agree that this sentence needed improvement. We have now specified that acetylcholine is necessary for contraction of skeletal, smooth and cardiac muscle, and as requested, have also included references to support the role of acetylcholine in brain development.

These are some examples to explain the necessary revision that I suggest to improve the manuscript.

Other suggestions:

- Page 1, line 31: "Whilst insecticides are an integral component of terrestrial ecosystems "

What intended the authors explain? Are the insecticides normally present in terrestrial ecosystems?

This was an error on our part; the sentence should have read “Whilst insects constitute an integral component of terrestrial ecosystems, their control requires significant expenditure from the agricultural sector”. This has now been corrected; the importance of insects to terrestrial ecosystems has now also been supported with a reference.

- In figure 1 for organophosphate insecticide (OP; red hexagons), OP is not reported.

We are not sure what this comment refers to; the red hexagons representing OP are present in the figure, in the pictorial key in the figure, and also in the figure legend.

- The conclusion is complex to read. I suggest revising the long sentences and correcting grammatical errors.

We agree that some of parts of the conclusion were overly complex. Some of the information, for example the  statistics related to crop yields following pesticide treatment, is unnecessary given that it had been presented elsewhere in the review. To address this, it has been removed to simplify the structure of the conclusion. 

Reviewer 2 Report

This paper is of very high quality, Authors comprehensively reviewed the consequences of OPs toxicity. The table is an advantage of this paper. Paper is ready for publication. In editing processs i would like to ask to remove some abbr. e.g. pns, nte, pbp etc. Its too many of them which makes reading a little bit difficult. Fig. 2: doubled "agricultural". Fig. 3: "short-term memory" what? i would add "short-term memory loss or impairment". Best wishes.

Author Response

This paper is of very high quality, Authors comprehensively reviewed the consequences of OPs toxicity. The table is an advantage of this paper. Paper is ready for publication.

We thank the reviewer for their very positive assessment of our work.

In editing processs i would like to ask to remove some abbr. e.g. pns, nte, pbp etc. Its too many of them which makes reading a little bit difficult.

While we agree that the use of many of these abbreviations may make reading difficult, the abbreviations mentioned are relatively common (particularly with regard to the peripheral nervous system; PNS and neuropathy-target-esterase; NTE) and necessary to avoid the repetition of the full term. We have, however removed abbreviations that were only used once (such as PBP, TNF, IL-6, etc.).

Fig. 2: doubled "agricultural".

This has been corrected.

Fig. 3: "short-term memory" what? i would add "short-term memory loss or impairment". Best wishes.

This has been corrected to “short term memory loss”.